Effects of growing Coptis chinensis Franch in the natural understory vs. under a manmade scaffold on its growth, alkaloid contents, and rhizosphere soil microenvironment

Wang Yu 1 2
Mo Yu R. 1 2
Tan Jun 1 2
Wu Li X. 1 2
Pan Yuan 1 2
Chen Xia D. 1 2 17837@163.com
1 Chongqing academy of Chinese Materia Medica , Chong Qing , China
2 Chongqing Subcenter of National Resource Center for Chinese Materia Medica, China Academy of Chinese Medical Science , Chongqing , China
Sotelo-Mundo Rogerio
Electronic publication date: 2022 Jul 20
Publication date: 2022
Volume: 10
Electronic Location ID: e13676
Received 2021 Nov 15; Accepted 2022 Jun 13
Copyright: © 2022 Wang et al.
Copyright year: 2022
Copyright holder: Wang et al.
License: This is an open access article distributed under the terms of the Creative Commons Attribution License, which permits unrestricted use, distribution, reproduction and adaptation in any medium and for any purpose provided that it is properly attributed. For attribution, the original author(s), title, publication source (PeerJ) and either DOI or URL of the article must be cited.
License URL: https://creativecommons.org/licenses/by/4.0/

Keywords: Coptis chinensis Franch, Growth and development, Soil physicochemical properties, Enzyme activities, Microbial communities

Funding: National Key Research and Development Program of China 2017YFC1702601 Basic Scientific Research Project of Chongqing Province jbky20190022 Natural Science Foundation of Chongqing Province cstc2019jcyj-msxmX0677 Basic Scientific Research Project of Chongqing Province No. cstc2019jxjl-jbky10004 This work was supported by the National Key Research and Development Program of China (No. 2017YFC1702601), the Basic Scientific Research Project of Chongqing Province (No. jbky20190022), the Natural Science Foundation of Chongqing Province (No. cstc2019jcyj-msxmX0677) and the Basic Scientific Research Project of Chongqing Province (No. cstc2019jxjl-jbky10004). The funders had no role in study design, data collection and analysis, decision to publish, or preparation of the manuscript.

==============================
Background

The main planting modes currently used for the production of Coptis chinensis Franch are under the shade of a manmade scaffold or a natural understory. In this study, we analysed changes in the growth, development, and alkaloids of C. chinensis when grown in a natural understory compared with under a manmade scaffold. We also clarified the differences in the rhizosphere soil microenvironment, represented by soil physicochemical factors, enzyme activity, and microbial community structure of 1- to 5-year-old C. chinensis between the different planting modes. These results will provide theoretical guidance and scientific evidence for the development, application, and extension of ecological planting technologies for C. chinensis.

Results

The results of this study showed that rhizome length, rhizome diameter, and rhizome weight all increased over time in both planting modes. The greatest rhizome length was reached in 4-year-old C. chinensis, while the greatest rhizome diameter and rhizome weight were obtained in 5-year-old C. chinensis. There was no significant difference in rhizome biomass between the two planting modes. The alkaloid content of the four common alkaloids in the rhizome of 5-year-old C. chinensis at the harvest stage met the standards found in the Pharmacopoeia of the People’s Republic of China; the berberine content and total alkaloids in the rhizomes were significantly higher with natural understory planting compared to planting under a manmade scaffold. A redundancy analysis revealed that the physicochemical factors and enzyme activity of rhizosphere soil were significantly correlated with variation in microbial community structure. Soil pH, available potassium, bulk density, available nitrogen, catalase, and peroxidase were all significantly correlated with bacterial and fungal community structures. Among these, soil pH was the most important factor influencing the structures of the fungal and bacterial community. In the two planting modes, the differences in soil enzyme activity and microbial community structure mainly manifested in the rhizosphere soil of C. chinensis between different growth years, as there was little difference between the rhizosphere soil of C. chinensis in a given growth year under different planting modes. The levels of nitrogen, phosphorus, potassium, and organic matter in the rhizosphere soil under either planting mode were closely associated with the type and amount of fertiliser applied to C. chinensis. Investigating the influence of different fertilisation practices on nutrient cycling in farmland and the relationship between fertilisation and the soil environment will be key to improving the yield and quality of C. chinensis medicinal materials while maintaining the health of the soil microenvironment.

Introduction

Rhizoma coptidis, a commonly used traditional Chinese medicine, is the dried rhizome of Coptis chinensis Franch, a plant species in the family Ranunculaceae. It was first documented in Divine Farmer’s Classic of Materia Medica (Shen Nong Ben Cao Jing) and is listed as the top grade. Rhizoma coptidis tastes cold and nurtures the heart, spleen, stomach, liver, gallbladder, and large intestine meridian. It clears away heat and dries dampness, in addition to purging intense heat and detoxification. Alkaloids are the main active ingredients of C. chinensis (Li et al., 2018). C. chinensis rhizomes are harvested 5 years after planting and are used in traditional Chinese medicine as a source for isoquinoline alkaloids. Generally, C. chinensis mainly consists of six alkaloids: berberine, coptisine, palmatine, jatrorrhizine, epiberberine, and columbamine (Wang et al., 2004). Among these, berberine (~7%), the dominant alkaloid, is known to have multiple beneficial physiological effects (Xie et al., 2004). These alkaloids also have various pharmacological properties including properties for treating infectious and inflammatory diseases (Zhang et al., 2011; Wu et al., 2016), as well as properties related to the prevention and treatment of health problems related to the nervous system and cardiovascular system, including diabetes and cancer (Zhen et al., 2011; Chou et al., 2017). Today, C. chinensis is mainly cultivated artificially and is produced in several regions, including Chongqing, Hubei, Sichuan, and Hunan. The Shizhu County of Chongqing is known as the “Hometown of C. chinensis” in China, as it alone produces more than 60% of China’s total crop of C. chinensis (Huang, 2018).

The main planting modes currently used for the production of C. chinensis are planting C. chinensis under the shade of a manmade scaffold or as natural understory. Planting C. chinensis under a manmade scaffold requires deforestation, which seriously damages the ecological environment. In contrast, planting C. chinensis in the natural understory takes advantage of the natural tree shade and thereby markedly reduces the labour required for planting, and is an environmentally friendly, ecological planting method worthy of promotion.

Studies of farmers and their farming practices in the C. chinensis production area of Shizhu have found that the yield of C. chinensis planted in the natural understory is approximately 20% lower compared to plants under a manmade scaffold. The current market price of C. chinensis is 150 Chinese yuan/kg. Based on the conventional yield of 200/kg per mu, the income per mu under a manmade scaffold is 6,000 Chinese yuan higher than that in natural understory. Although planting C. chinensis in the natural understory is simpler to do, it is still difficult to promote this planting mode on a large scale due to the much lower income yield.

Soil microorganisms make up one of the most active components of soil. They influence the soil environment by participating in soil formation, organic matter decomposition, and soil nutrient transformation (Miya & Firestone, 2000; Zhang et al., 2017; Garcia-Ruiz et al., 2008). Soil enzymes are the secretions of live microorganisms, animals, and plants, or a class of biologically active substances with catalytic ability that are released into the soil through the decomposition of animal and plant residues and remains (Zhang et al., 2007; Liu et al., 2008; Wang et al., 2013). The available and total nutrients in rhizosphere soil, the enzyme activity associated with soil fertility, and the number and type of microorganisms are vital indicators of soil quality and fertility (Velasquez, Lavelle & Andrade, 2007).

In this study, we analysed the differences in the growth and development of C. chinensis, as well as in the soil microenvironment, including rhizosphere soil nutrient conditions, enzyme activity, and microbial community structure in natural understory planting compared to planting under a manmade scaffold. The effects of the two planting modes on C. chinensis plant growth were clarified to provide theoretical guidance and scientific evidence for the development, application, and extension of ecological planting technologies for C. chinensis.

Materials and Methods

Sample collection

In November 2018, soil samples were collected at the C. chinensis Good Agricultural Practices base in Shizhu County, Chongqing, China. Table 1 shows the geographical environment of the C. chinensis planting area. To eliminate differences in the soil environment caused by original soil differences and varying fertilisation and management methods, we selected adjacent C. chinensis plots growing under a manmade scaffold and in the natural understory, which were planted by the same household. Based on an interview and investigation, the sampling locations were finally selected from the Maolaocheng Group in Yangdong Village, Huangshui Town, Shizhu County. The same cultivation and management methods were used for growing C. chinensis in the natural understory and under a manmade scaffold. A total of 150 kg/ha of urea and 1,500 kg/ha of superphosphate were applied to 1-year-old C. chinensis. A total of 150 kg/ha of urea, 150 kg/ha of ammonium bicarbonate and 3,000 kg/ha of superphosphate were applied to 2-year-old C. chinensis, and 450 kg/ha of ammonium bicarbonate and 3,000 kg/ha of superphosphate 3,000 kg/ha were applied to 3- and 4-year-old C. chinensis. No topdressing was applied in the harvest year of 5-year-old C. chinensis at the harvest stage.

Table 1 Geographical environment of the C. chinensis planting area.

	Shading ratio (%)	Slope
(°)	Altitude
(m)	Longitude	Latitude	Soil bulk density
(g/cm3)	pH	
MD1	83	20	1,561	30°18′01″	108°25′43″	0.61	4.08	
MD2	76	18	1,561	30°18′01″	108°25′43″	0.95	4.50	
MD3	80	40	1,567	30°18′11″	108°25′39″	0.72	3.98	
MD4	82	28	1,567	30°18′02″	108°25′37″	0.68	3.88	
MD5	13	30	1,560	30°18′01″	108°25′42″	1.00	4.09	
ML1	90	32	1,662	30°18′16″	108°25′29″	4.32	0.58	
ML2	80	32	1,662	30°18′16″	108°25′29″	4.00	0.75	
ML3	83	46	1,568	30°18′01″	108°25′41″	4.01	0.74	
ML4	42	40	1,662	30°18′16″	108°25′29″	3.69	0.61	
ML5	32	46	1,553	30°18′00″	108°25′41″	3.79	0.87	
Note:

ML1–ML5: 1–5-year-old C. chinensis in natural understory; MD1–MD5: 1–5-year-old C. chinensis on manmade scaffold. The same below.

When sampling C. chinensis, 10 similarly-sized plants with normal growth were selected at random. The cutting-ring method was used to collect soil samples to determine soil bulk density (Yi, 2007).

Then, rhizosphere soil samples were collected from the soil of growing C. chinensis plants using a five-point method (Shao, Wang & Huang, 2006). Four soil samples were collected from each plot as replicates. Each C. chinensis plant with its whole root system was taken and gently shaken to remove root-zone soil. The soil attached to the root surface was collected as rhizosphere soil and immediately placed in a refrigerated vehicle at −10 °C. The soil samples were brought to the laboratory and stored in two parts: one part was stored at −80 °C for later analyses of soil enzyme activity and microbial communities; the other part was air-dried, passed through a 2-mm sieve, and stored at room temperature to determine its physicochemical properties.

Analytical methods

Plant morphology and biomass measurement

After retrieving all the C. chinensis sample plants, plant height was measured. The plants were separated into different growth parts: leaves, rhizomes, and fibrous roots. The rhizomes and fibrous roots were washed, followed by measurements of rhizome length and diameter and fibrous root length. The weight of each plant part was measured. Finally, the rhizomes were dried in a 55 °C oven to determine their alkaloid contents.

Alkaloid content analysis

Levels of coptisine, berberine, epiberberine, palmatine hydrochloride, jatrorrhizine hydrochloride, columbamine, and magnoflorine were measured in dry rhizomes (Wang et al., 2018). Chromatographic conditions were as follows: chromatographic column, Phenomenex-NX; mobile phase A = acetonitrile–30 mmol·L−1 ammonium bicarbonate (containing 0.7% ammonia and 0.1% triethylamine); mobile phase B = acetonitrile; column temperature, 30 °C; flow rate, 1 mL·min−1; and detection wavelength, 270 nm.

Light intensity measurement

At noon on a sunny day, light intensity was measured five times for C. chinensis plants in each growth year, and the mean value was taken. The instrument was placed 10 cm directly above the plant during measurement. Light intensity was simultaneously measured in an unshaded place as a control, and this measurement was then used to calculate the shading ratio.

Soil physicochemical analysis

Soil bulk density was measured using the cutting-ring method. Soil pH was measured using a pH meter (water/soil = 1:2.5). Soil organic matter content was determined using the potassium dichromate oxidation–volumetric method. Total nitrogen was analysed using the Kjeldahl method after sulfuric acid–accelerator digestion. Available nitrogen was quantified using the alkaline hydrolysis diffusion method. Available phosphorus was determined by the molybdenum antimony colorimetric method after extraction with 0.5 M sodium bicarbonate. Available potassium was determined by atomic-absorption spectrophotometry after extraction with 1 M ammonium acetate. Total phosphorus, potassium, and nitrogen levels were determined by inductively coupled plasma–atomic emission spectrometry. All the analytical methods used are those referenced in Analytical Methods of Soil Agro-chemistry (Lu, 2000).

Soil enzyme activity assays

Soil urease activity, polyphenol oxidase activity, catalase activity, invertase activity, peroxidase activity, dehydrogenase activity, neutral phosphatase activity, and neutral protease activity were assayed using commercial kits (Suzhou Comin Biotechnology Co., Ltd., Suzhou, China).

Soil microbial analysis

Genomic DNA extraction, amplification, and high-throughput sequencing

The cetyltrimethylammonium bromide method was used to extract genomic DNA from soil samples. Then, agarose gel electrophoresis was conducted to determine the purity and concentration of each extracted DNA. An appropriate amount of sample DNA was transferred into a centrifuge tube and diluted to 1 ng/μl with sterile water. The diluted genomic DNA served as a template for the polymerase chain reaction (PCR), which was carried out using specific primers with barcodes according to the sequencing region targeted: Phusion® High-Fidelity PCR Master Mix with GC Buffer (New England Biolabs, Ipswich, MA, USA), and high-efficiency high-fidelity DNA polymerase to ensure high amplification efficiency and accuracy. The bacterial primer-targeted region was 16S rRNA gene V4 region (primers 515F and 806R) and the fungal primer-targeted region was ITS rRNA gene ITS1 region (primers ITS5-1737F and ITS2-2043R). The PCR products were diluted to equal concentrations and then separated by 2% agarose gel electrophoresis in 1× TAE buffer. The target bands were cut and recovered. The excised PCR products were purified using the GeneJET gel extraction kit (Thermo Scientific, Waltham, MA, USA). Libraries were constructed using the Ion Plus Fragment Library Kit 48 rxns (Thermo Fisher, Waltham, MA, USA). After Qubit quantification and library qualification, the constructed libraries were sequenced on the Ion S5™XL platform (Thermo Fisher, Waltham, MA, USA).

Processing and species annotation of high-throughput sequencing data

Cutadapt (Aßhauer et al., 2015) was first used to cut the low-quality part of the reads. The data from different samples were then separated from the obtained reads according to barcode. Barcode and primer sequences were removed through preliminary quality control to obtain raw reads. After that, the raw reads needed to be processed to remove chimaera sequences. The reads were compared with the species annotation database to detect chimaeras using VSEARCH (https://github.com/torognes/vsearch/; Martin, 2011), and chimaeras were finally removed (Rognes et al., 2016) to obtain clean reads. The clean reads of all samples were clustered using Uparse (Haas et al., 2011). By default, sequences with at least 97% identity were clustered into operational taxonomic units (OTUs). Representative sequences of the OTUs were selected according to the principle of the algorithm; that is, the sequence with the highest frequency in each OTU was selected as the representative sequence of that OTU. The Silva Database (Kõljalg et al., 2013) was used based on the Mothur algorithm to annotate the taxonomic information of bacterial sequences. The Unite Database (Quast et al., 2013) was used based on the Blast algorithm, which was calculated by QIIME software (Version 1.9.1) to annotate the taxonomic information of the fungal sequences, and the microbial community composition of samples was analysed at each taxonomic level. Rapid multiple-sequence alignment was done in MUSCLE version 3.8.31 (Quast et al., 2013) to obtain the phylogenetic relationship of all OTUs. Finally, the data from all samples were normalised to the sample with the smallest data size.

Environmental factor correlation analysis

A redundancy analysis (also called principal components analysis of instrumental variables) is a technique used to analyse two sets of variables, one set being dependent on the other. In our redundancy analysis, the envfit function was used to find the influence of each environmental factor on genus distribution. Similarly, the environmental factors or combinations most associated to the correlation of the species were screened by the bioenv function in R. A variance partial analysis (VPA) is a method which uses the rda (X,Y,Z) function of the vegan package to quantify the amount of species distribution explained by each environmental factor to help us understand the effect of the main environmental factor (Y) and the coenvironmental factor (Z) on species distribution.

Results

C. chinensis plant growth and development under two planting modes

We summarize the plant growth and development of C. chinensis under the two planting modes in Table 2. In the manmade scaffold planting mode, the above-ground plant height, leaf number, leaf blade weight, and petiole weight of C. chinensis all grew over time to reach their maximum values in 4-year-old C. chinensis, then decreased significantly in 5-year-old C. chinensis. In the natural understory planting mode, the above-ground plant height, leaf number, and leaf blade weight of C. chinensis all grew over time and reached their maximum values in 5-year-old C. chinensis. Under both planting modes, rhizome length, rhizome diameter, and rhizome weight all exhibited an upward trend with time; the greatest rhizome length was in 4-year-old C. chinensis, while the greatest rhizome diameter and rhizome weight were reached in 5-year-old C. chinensis.

Table 2 Dynamic changes in the growth of Coptis chinensis under manmade scaffold and natural understory planting modes.

	Plant height
(cm)	Leaf number
(cm)	Leaf blade weight
(g)	Petiole
weight
(g)	Fibrous root length
(cm)	Fibrous root weight
(g)	Rhizome length
(cm)	Rhizome diameter
(cm)	Rhizome weight
(g)	
MD1	7.30 ± 0.949e	8.90 ± 4.383ef	1.18 ± 0.397c	0.53 ± 0.163e	3.70 ± 0.949d	1.55 ± 0.856e	2.40 ± 0.682c	4.59 ± 1.326d	0.71 ± 0.345e	
MD2	10.60 ± 2.011cd	23.40 ± 8.343cd	4.37 ± 1.320b	1.82 ± 0.689cde	6.60 ± 1.956bc	4.84 ± 2.463cde	5.41 ± 0.853b	6.67 ± 1.695c	4.12 ± 2.004de	
MD3	15.90 ± 3.843ab	26.70 ± 8.500cd	4.31 ± 1.790b	2.95 ± 1.122bcd	8.20 ± 1.398ab	5.18 ± 2.290bcde	6.14 ± 1.391ab	7.71 ± 0.422bc	7.71 ± 2.534cd	
MD4	16.20 ± 2.098a	44.90 ± 21.937a	8.64 ± 5.217a	7.98 ± 5.925a	6.20 ± 2.394bc	9.78 ± 6.019abc	6.99 ± 0.832a	8.28 ± 1.663bc	20.19 ± 10.248a	
MD5	14.89 ± 5.232ab	34.50 ± 18.180abc	6.01 ± 3.637ab	4.72 ± 3.028b	6.80 ± 2.098bc	10.08 ± 9.474ab	6.60 ± 1.132ab	9.06 ± 2.457ab	21.87 ± 13.901a	
ML1	8.55 ± 2.315de	6.70 ± 2.497f	0.44 ± 0.310c	0.21 ± 0.134e	5.05 ± 0.762cd	0.41 ± 0.152e	1.58 ± 0.349c	4.23 ± 0.861d	0.24 ± 0.084e	
ML2	13.00 ± 3.334bc	20.50 ± 6.519de	4.72 ± 2.352b	1.85 ± 0.779cde	9.20 ± 1.135a	3.89 ± 1.487de	6.37 ± 0.996ab	7.29 ± 1.646c	5.49 ± 2.270cde	
ML3	13.22 ± 4.216abc	30.60 ± 16.036bcd	7.67 ± 2.881a	3.24 ± 1.069bc	9.20 ± 2.821a	13.26 ± 5.068a	6.04 ± 1.883ab	7.91 ± 1.501bc	10.81 ± 2.535bc	
ML4	14.40 ± 3.701ab	40.22 ± 18.444ab	6.59 ± 4.058ab	5.41 ± 3.007bde	6.33 ± 3.500bc	10.08 ± 8.805ab	6.84 ± 1.498a	9.32 ± 2.427ab	15.79 ± 7.624ab	
ML5	14.30 ± 3.434ab	45.10 ± 16.455a	7.15 ± 3.593ab	4.80 ± 3.147b	8.20 ± 2.348ab	7.10 ± 5.023bcd	6.43 ± 1.543ab	10.17 ± 1.992a	18.68 ± 7.962a	
Note:

Different letters in the same column mean a significant difference at the 0.05 level.

We then conducted an analysis of variance (ANOVA) on the above-ground plant height, leaf number, leaf blade weight, and petiole weight of 5-year-old C. chinensis under the two planting modes. No significant differences were observed in plant height or petiole weight, but leaf number and leaf blade weight were significantly higher in the natural understory plants than in the plants grown under a manmade scaffold. As the rhizome of this species is medicinal, the underground rhizome diameter, rhizome length, and rhizome weight of C. chinensis are the three indicators that determine the yield of the medicinal material, so we compared all three in the same growth years between the two planting modes by ANOVA. Rhizome length showed a significant difference in 1-year-old and in 2-year-old C. chinensis plants, but otherwise, there were no significant differences in the rhizome parameters of C. chinensis plants in a given growth year between the two planting modes. ANOVA also indicated no significant difference in the drying rate of rhizomes from 5-year-old C. chinensis in the natural understory (39.76%) from the ones under a manmade scaffold (41.09%).

Alkaloid content of C. chinensis rhizomes under the two planting modes

In 2015, the Pharmacopoeia of the People’s Republic of China outlined the required levels of the four alkaloids (berberine, epiberberine, coptisine, and palmatine) in the rhizomes of C. chinensis needed for medicinal use. We found that the contents of all four alkaloids in the rhizomes of 2- to 5-year-old C. chinensis under both planting modes (Table 3) met these requirements (Chinese Pharmacopoeia Commission, 2015). We next compared the alkaloid content in the rhizomes of C. chinensis in the same growth years under the two different planting modes. In the rhizomes of 2-year-old C. chinensis, there were no significant differences in epiberberine, coptisine, jatrorrhizine, or magnoflorine content, but berberine, palmatine, columbamine levels as well as total alkaloid content were significantly higher under a manmade scaffold than in the natural understory. There was no significant difference in the alkaloid content in the rhizomes of 3-year-old C. chinensis between the different planting modes, and only magnoflorine and columbamine showed significant differences between groups in the rhizomes of 4-year-old C. chinensis with both showing higher levels in the rhizomes under a manmade scaffold. In 5-year-old C. chinensis at the harvest stage, berberine and magnoflorine levels and total alkaloid content in the rhizomes were significantly higher in the natural understory than under a manmade scaffold, whereas epiberberine, palmatine, and columbamine contents were all significantly higher under a manmade scaffold than in the natural understory.

Table 3 Alkaloid content in Coptis chinensis rhizomes under the two planting modes (%).

	Berberine	Epiberberine	Coptisine	Palmatine	Jatrorrhizine	Magnoflorine	Columbamine	Total alkaloids	
MD2	7.04 ± 0.128a	5.42 ± 0.127a	1.69 ± 0.038de	1.85 ± 0.034a	0.79 ± 0.006c	0.37 ± 0.006c	1.16 ± 0.020a	18.31 ± 0.329a	
MD3	5.78 ± 0.037d	4.93 ± 0.047c	1.93 ± 0.020a	1.71 ± 0.012c	0.85 ± 0.007a	0.36 ± 0.008c	0.95 ± 0.003bc	16.52 ± 0.099c	
MD4	6.31 ± 0.034c	4.73 ± 0.029d	1.94 ± 0.019a	1.74 ± 0.009b	0.81 ± 0.014bc	0.37 ± 0.021c	1.10 ± 0.012a	16.99 ± 0.106b	
MD5	5.89 ± 0.063d	5.11 ± 0.065b	1.76 ± 0.021cd	1.60 ± 0.017e	0.79 ± 0.016c	0.38 ± 0.039c	1.00 ± 0.013b	16.54 ± 0.160c	
ML2	5.39 ± 0.012e	5.30 ± 0.027a	1.63 ± 0.015e	1.66 ± 0.005d	0.79 ± 0.012c	0.38 ± 0.011c	0.98 ± 0.007bc	16.13 ± 0.038d	
ML3	5.80 ± 0.018d	4.85 ± 0.067cd	1.88 ± 0.129ab	1.70 ± 0.021c	0.85 ± 0.003a	0.35 ± 0.004c	0.93 ± 0.059bc	16.37 ± 0.085cd	
ML4	6.27 ± 0.133c	4.82 ± 0.143cd	1.90 ± 0.079ab	1.73 ± 0.006bc	0.83 ± 0.020b	0.41 ± 0.006b	0.94 ± 0.112bc	16.91 ± 0.271b	
ML5	6.51 ± 0.056b	4.82 ± 0.002cd	1.82 ± 0.024bc	1.55 ± 0.010f	0.79 ± 0.014c	0.51 ± 0.003a	0.90 ± 0.001c	16.90 ± 0.106b	
Note:

Different letters in the same column mean a significant difference at the 0.05 level.

Physicochemical properties of rhizosphere soil of C. chinensis under the two planting modes

In the manmade scaffold planting mode, the organic matter, total nitrogen, available nitrogen, total phosphorus, available phosphorus, and total potassium contents were all lowest in the rhizosphere soil of 1-year-old C. chinensis (Fig. 1). These nutrient contents increased over time and peaked in the rhizosphere soil of 4-year-old C. chinensis, followed by a significant decrease in the rhizosphere soil of 5-year-old C. chinensis. The available potassium content gradually increased with time, peaking in the rhizosphere soil of 5-year-old C. chinensis. In the natural understory planting mode, the lowest organic matter, total nitrogen, available nitrogen, total phosphorus, and available phosphorus contents were observed in the rhizosphere soil of 1-year-old C. chinensis. These nutrient contents increased gradually with time to reach their peaks in the rhizosphere soil of 3- to 4-year-old C. chinensis, followed by a significant decrease in 5-year-old C. chinensis. However, the total potassium and available potassium were relatively high in the rhizosphere soil of 1-year-old C. chinensis and continuously changed with the growth and development of C. chinensis. In the fifth year, the total potassium and available potassium were significantly lower in the rhizosphere soil than they were one year after planting.

Figure 1 Physicochemical properties of rhizosphere soil of Coptis chinensis under both planting modes.

Different lowercase letters were used in these figures mean a significant difference at the 0.05 level.

We also analysed the nutrient composition of rhizosphere soil for C. chinensis in the same growth years under different planting modes using ANOVA. The available nitrogen, total phosphorus, available phosphorus, total potassium, available potassium, and organic matter levels, but not total nitrogen level, were all significantly higher in the soil of 1-year-old C. chinensis in the natural understory than in the soil of the plants under a manmade scaffold. In the soil of 2-year-old C. chinensis in the natural understory, except for total potassium and organic matter, the contents of all the nutrients were higher than in the soil of those under a manmade scaffold. In the soil of 3-year-old C. chinensis in the natural understory, except for available potassium and organic matter, the contents of all the nutrients were higher than in the soil of those under a manmade scaffold. In the soil of 4-year-old C. chinensis in the natural understory, the organic matter, available potassium, and available phosphorus contents were significantly higher while total nitrogen, available nitrogen, total phosphorus, and total potassium levels were significantly lower than they were in the soil under a manmade scaffold. In the soil of 5-year-old C. chinensis, organic matter and available nitrogen levels were significantly higher in the natural understory, while available potassium and available phosphorus were significantly higher under a manmade scaffold, and there was no significant difference in total nitrogen, total phosphorus and total potassium between the two planting modes.

Enzyme activity in rhizosphere soil of C. chinensis under both planting modes

We analysed the enzyme activity in the rhizosphere soil of C. chinensis between different growth years in the same planting mode by ANOVA (Fig. 2). In the natural understory planting mode, dehydrogenase activity and invertase activity in the rhizosphere soil of C. chinensis gradually increased with time. Urease activity in the rhizosphere soil of 1-year-old C. chinensis was significantly higher than that of C. chinensis in any other growth year, whereas peroxidase activity decreased gradually with time. Neutral protease activity and neutral phosphatase activity both showed dynamic changes, first decreasing and then increasing until there were no significant differences in the enzyme activity of the rhizosphere soil between the 5- and 1-year-old C. chinensis plants. There were no significant differences in the polyphenol oxidase activity and catalase activity in the rhizosphere soil of C. chinensis between different growth years. In the manmade scaffold planting mode, invertase activity increased gradually over time, while catalase activity gradually decreased over time. Polyphenol oxidase activity, peroxidase activity, and neutral protease activity first decreased and then increased with time, until no significant differences were observed in the enzyme activity of the rhizosphere soil between the 5- and 1-year-old C. chinensis plants. Dehydrogenase activity and neutral phosphatase activity did not differ significantly in the rhizosphere soil of C. chinensis among different growth years.

Figure 2 Changes in enzyme activity in rhizosphere soil of Coptis under both planting modes.

Different lowercase letters were used in these figures mean a significant difference at the 0.05 level.

Next, we compared the enzyme activity in the rhizosphere soil of C. chinensis in a given growth year between the two planting modes. The differences in enzyme activity were mainly found in the rhizosphere soil of 1- and 2-year-old C. chinensis. Catalase activity in the rhizosphere soil of 1- and 2-year-old C. chinensis was significantly higher under a manmade scaffold than in the natural understory. Peroxidase activity in the rhizosphere soil of 2-year-old C. chinensis in the natural understory was significantly higher than under a manmade scaffold, but urease activity in the rhizosphere soil of 2-year-old C. chinensis was significantly higher under a manmade scaffold. No significant differences were observed in polyphenol oxidase, dehydrogenase, neutral protease, invertase, or neutral phosphatase activity in the rhizosphere soil of C. chinensis in the same growth year in the natural understory vs. under a manmade scaffold.

The above results indicate that the differences in enzyme activity were mainly found in the rhizosphere soil of C. chinensis between different growth years, while little difference occurred in the enzyme activity of rhizosphere soil of C. chinensis in the same growth years under the two different planting modes.

Composition and relative abundance of dominant microbial communities

The fungal taxa identified at levels >1% in the rhizosphere soil of 1- to 5-year-old C. chinensis were identical at the class level between the two planting modes, though there were differences in the levels of the fungal taxa identified. The following fungal taxa were identified at levels >1%: Mortierellomycetes, Agaricomycetes, Sordariomycetes, Leotiomycetes, Dothideomycetes, Archaeorhizomycetes, Tremellomycetes, Eurotiomycetes, and unidentified_Ascomycota (Fig. 3).

Figure 3 Relative abundance of the major fungal communities at the class level.

As with the fungal taxa identified, the bacterial taxa identified with levels >1% in the rhizosphere soil of 1- to 5-year-old C. chinensis were identical at the class level between the two planting modes (Fig. 4). The bacterial classes identified at levels >1% were: Alphaproteobacteria, Gammaproteobacteria, Acidobacteria, unidentified_Actinobacteria, Deltaproteobacteria, Bacteroidia, Planctomycetacia, Verrucomicrobiae, Thermoleophilia, Acidimicrobiia, unidentified_Gemmatimonadetes, Ktedonobacteria, and unidentified_Cyanobacteria.

Figure 4 Relative abundance of the major bacterial communities at the class level.

Microbial alpha-diversity indices

The alpha-diversity indices represented by observed species, Shannon’s index, and the Chao1 index indicate species richness and the diversity and evenness of the species distribution in a sample (Li et al., 2013). We compared the three diversity indices between groups and graphed the significant differences in microorganisms between them (Figs. 5 and 6). The comparison of alpha-diversity indices in the rhizosphere soil of C. chinensis between different modes showed that there were no significant differences in the observed species, Chaol, or Shannon index of fungi in the rhizosphere soil between the two modes. The Chaol and Shannon indices of bacteria in the rhizosphere soil of C. chinensis also showed no significant differences, but the observed species index of bacteria in the rhizosphere soil of 1- and 2-year-old C. chinensis was significantly lower in the natural understory planting mode than under a manmade scaffold. The differences in alpha-diversity indices were mainly reflected in the rhizosphere soil of C. chinensis between different growth years, and there was little difference in the rhizosphere soil of C. chinensis in a given growth year between the two planting modes. The alpha-diversity indices of bacteria had more pronounced differences, while the alpha-diversity indices of fungi had minor differences in the rhizosphere soil of C. chinensis in different growth years.

Figure 5 Alpha diversity indices of fungi.

Figure 6 Alpha diversity indices of bacteria.

Microbial species differences between groups

We identified the differential microorganisms at the species level in the rhizosphere soil of C. chinensis in the same growth years under the two different planting modes using the Student’s T-test (significance level p < 0.05). The comparison of fungal species between the two groups showed that the level of Solicoccozyma terricola in the rhizosphere soil of 2-year-old C. chinensis in the natural understory was significantly higher than that under a manmade scaffold, while the level of Fusarium oxysporum was significantly higher in the rhizosphere soil of C. chinensis under a manmade scaffold than in the natural understory (Fig. 7). The level of Solicoccozyma terricola in the rhizosphere soil of 3-year-old C. chinensis under a manmade scaffold was significantly higher than that in the natural understory, while the level of Chytridium olla in the rhizosphere soil of C. chinensis in the natural understory was significantly higher than that under a manmade scaffold.

Figure 7 Significance test of species difference between fungal groups.

The comparison of bacterial species between groups found that bacterium Ellin5102, bacterium Ellin6510, Rhodospirillaceae bacterium L34 and bacterium Ellin6526 were significantly more abundant in the rhizosphere soil of 1-year-old C. chinensis under a manmade scaffold than in the natural understory (Fig. 8). In the rhizosphere soil at 2 years, Rhodanobacter sp., Bacterium Ellin7505, bacterium Ellin5102, bacterium Ellin6089, and Rhizobiales bacterium GAS188 were significantly more abundant in the natural understory, while beta proteobacterium WF17, bacterium Ellin6067, and alpha proteobacterium BAC29 were significantly more abundant under a manmade scaffold. In the rhizosphere soil at 3 years, Bacterium Ellin6543, Collimonas arenae and bacterium Ellin6100 were more abundant in the natural understory, while bacterium Ellin6515 and actinobacterium BGR88 were more abundant under a manmade scaffold. In the rhizosphere soil of 4-year-old C. chinensis, Gemmatimonadetes_bacterium_WY71 was more abundant under a manmade scaffold. There were no significant differences in bacterial species in the rhizosphere soil of 5-year-old C. chinensis between the two different planting modes.

Figure 8 Significance test of species difference between bacterial groups.

Based on the above analysis, we found that the differences in microbial species in the rhizosphere soil of C. chinensis between the two different planting techniques were mainly in the levels of the microbial species present rather than the diversity of species, as we found no species unique to one planting mode. Differences in soil fungi only appeared in the rhizosphere soil of 2- and 3-year-old C. chinensis, and only three species had significantly different levels between the two planting groups. Compared with fungi, soil bacteria levels showed larger differences with a total of 17 bacterial species showing significant differences in abundance levels between the two planting groups. The differences in abundance levels of bacteria showed dynamic changes over time, gradually increasing, then decreasing, and in the end showing no difference from the 1-year values.

Correlations between microbial community structure and soil physicochemical factors

The distance-based redundancy analysis (RDA) is currently the most widely used environmental factor analysis in the field of microbiology. It is mainly used to indicate the relationship between microbial communities and environmental factors. The RDA analysis of the soil fungal community structure and soil physicochemical factors in this study showed that soil pH and bulk density were positively correlated with variation in fungal community structure, while available nitrogen and available potassium were negatively correlated with variation in fungal community structure (Fig. 9). Their correlation levels were ranked as follows: pH > available potassium > bulk density > available nitrogen. The RDA analysis of bacteria showed that among the physicochemical factors in the rhizosphere soil of C. chinensis, all indicators except for total potassium had a significant influence on variation in bacterial community structure (Fig. 10). Soil physicochemical factors ranked based on their influence on the soil bacterial community structure are as follows: pH > available phosphorus > total phosphorus > available potassium > bulk density > total nitrogen > available nitrogen > organic matter.

Figure 9 Redundancy analysis on physical-chemical properties and soil dominant fungal phylum.

Figure 10 Redundancy analysis on physical-chemical properties and soil dominant bacterial phylum.

Correlations between microbial community structure and soil enzyme activity

The RDA results of soil fungal community structure and soil enzyme activity showed that polyphenol oxidase activity, peroxidase activity, neutral protease activity, and urease activity were all negatively correlated with variation in fungal community structure, while catalase activity was positively correlated with variation in fungal community structure (Fig. 11). Their correlation levels were ranked as follows: invertase > neutral protease > peroxidase > urease > polyphenol oxidase. The RDA analysis of bacteria showed that dehydrogenase activity, catalase activity, and invertase activity were positively correlated with variation in bacterial community structure, while peroxidase activity and neutral phosphatase activity were negatively correlated with variation in bacterial community structure (Fig. 12). Their correlation levels were ranked as follows: invertase > dehydrogenase > catalase > peroxidase > neutral phosphatase.

Figure 11 Redundancy analysis on enzyme activity and soil dominant fungal phylum.

Figure 12 Redundancy analysis on enzyme activity and soil dominant bacterial phylum.

Contribution of soil environmental factors to microbial community variation

A variance partial analysis (VPA) can be used to identify the contribution of each environmental factor on the distribution of microbial communities. In this study, soil physicochemical factors represented by pH, bulk density, organic matter, total nitrogen, total phosphorus, total potassium, available nitrogen, available potassium, and available phosphorus were bundled as environmental factor env1; soil enzyme activity represented by polyphenol oxidase, dehydrogenase, catalase, invertase, neutral phosphatase, urease, peroxidase, and neutral protease were bundled as environmental factor env2. The percentage of microbial community variation in the rhizosphere soil of C. chinensis explained by the two groups of soil environmental factors—physicochemical factors and enzyme activity—was analysed. The results showed that soil physicochemical factors contributed to 33.20% of the fungal community variation, soil enzyme activity contributed to 25.80% of the variation, and their interaction contributed very little (only 0.19%) to the fungal community variation. Soil physicochemical factors contributed to 27.29% of the bacterial community variation, soil enzyme activity contributed to 14.98% of the bacterial community variation, and their interaction contributed to 29.22% of the bacterial community variation. Compared with soil enzyme activity, soil physicochemical factors had a larger influence on microbial community variation in the rhizosphere soil of C. chinensis. Bacterial communities were more influenced by soil environmental factors than fungal communities were. Overall, soil environmental factors, represented by soil physicochemical factors and enzyme activity contributed to 71.49% of the bacterial community variation and 59.18% of the fungal community variation (Fig. 13).

Figure 13 Variance partitioning canonical correspondence analysis between microbial community and soil environmental factors.

(A) Variance partitioning canonical correspondence analysis between fungus and soil environmental factors. (B) Variance partitioning canonical correspondence analysis between bacteria and soil environmental factors.

Discussion

Soil microorganisms are the drivers of soil organic matter transformation and nutrient cycling; they participate in processes related to the decomposition of soil organic matter and the transformation of nitrogen and phosphorus nutrients, in addition to regulating the energy balance enzyme activity of the soil (Zhang & Nan, 2010). A study that quantified soil microorganisms and enzyme activity in a maize interplanting system found that bacteria, fungi, and actinomycetes play a positive role in enhancing enzyme activity in soil (Zhang et al., 2012). In the present study, we also demonstrated that the physicochemical factors and enzyme activity in the rhizosphere soil of C. chinensis were significantly correlated with variation in microbial community structure. Soil pH, available potassium, bulk density, available nitrogen, catalase, and peroxidase all had significant correlations with bacterial and fungal community structures (Figs. 9–12). Among these, soil pH was the most important factor influencing variation in both fungal and bacterial community structures. Soil pH is a key factor of soil microbial diversity and community composition, with one study suggesting that pH has a universal influence on the distribution of microorganisms (Shen et al., 2013). Other studies have found negative correlations between the number of fungi present and some soil enzyme activity under continuous planting of peanut and watermelon, respectively (Zhao et al., 2008; Sun et al., 2001). A plausible explanation is that crop types, cultivation modes, and chemical fertilisation could change the pH values of soil and thereby supress soil enzyme activity.

In this study, in both planting modes, organic matter, total nitrogen, available nitrogen, total phosphorus, and available phosphorus contents in the rhizosphere soil of C. chinensis in different growth years all followed the same trend (Fig. 2): the nutrient contents were lowest in the rhizosphere soil of 1-year-old C. chinensis and increased gradually with time. Their maximum values were reached in the rhizosphere soil of 3- and 4-year-old C. chinensis, after which the values decreased significantly at the 5-year mark. These changes could be related to the type and amount of fertiliser given. In the 1st year of transplanting, C. chinensis plants are small, and the fertilisation rate is the lowest. With continuous growth of C. chinensis plants, the fertilisation rate is increased year by year to reach its highest value in the 4th year. Because C. chinensis is harvested in the 5th year, no topdressing is applied in this harvest year. As a result, the contents of various nutrients in the soil tend to decrease during this final year. This result indicates that fertiliser input is a key factor influencing the soil physicochemical properties of C. chinensis. Fertilisation can directly affect the physical properties and chemical compositions of the soil, cause different degrees of nutrient limitation on soil microorganisms (Wang & Yu, 2008), and thus influence the soil microbial community structure as well as enzyme activity in the soil (Xu et al., 2010; Lu et al., 2015; Li et al., 2012). Here, we found that the differences in soil enzyme activity and microbial community structure were mainly found between the rhizosphere soil of C. chinensis between different growth years, as there were only small differences between the rhizosphere soil of C. chinensis in the same growth year in the different planting modes. These results may be attributed to a few different factors. First, the types and amounts of fertiliser applied were mainly based on the growth and development of C. chinensis plants, so they changed considerably from year to year. In contrast, the fertiliser types and fertilisation rates for C. chinensis plants in the same growth year were exactly the same in the two planting modes. Therefore, the differences in fertilisation led to variation in the soil microenvironment of C. chinensis in different growth years. Second, root exudates are a key factor in the different rhizosphere microecological characteristics of plants (Wu, Lin & Lin, 2014), and the rhizosphere has active selectivity for soil microorganisms under the action of root exudates (Kowalchuk et al., 2002; Ziegler et al., 2013). For different plants and plants at different stages of growth and development, their rhizosphere exudates contain various secondary metabolites. These plant rhizosphere exudates have indirect ecological effects, inhibiting or promoting the growth of certain groups of microorganisms, which eventually have a selective effect on soil microbial communities (Haichar et al., 2008; Hartman et al., 2009). Therefore, C. chinensis plants may have unique rhizosphere microenvironments at different developmental stages.

Fertilisation is one of the largest factors in soil quality and sustainable farming (Hai et al., 2010). Long-term chemical fertilisation combined with organic fertiliser distinctly increases soil organic matter, nitrogen, phosphorus and potassium nutrient contents and promotes microbial metabolism and propagation, which creates a favourable soil ecological environment for a high, stable crop yield (Sun et al., 2004). Long-term combined application of organic and inorganic fertilisers markedly increases soil microbial biomass carbon and nitrogen contents in rice paddies to levels higher than those under the sole application of inorganic fertiliser or straw application treatments (Xu et al., 2016). This may be because the combined application of organic and inorganic fertilisers increases root biomass and root exudates, promoting the growth and reproduction of soil microorganisms. Another study found that either a deficiency in soil elements or an imbalance in fertilisation has a negative effect on the diversity of microbial communities. Soil sickness due to continuous planting and root rot disease, closely related to the soil environment, have become the largest factors limiting the development of C. chinensis (Wu et al., 2020). There is still no effective control method for root rot disease in C. chinensis, which can cause low crop yield or even no viable crop to harvest, seriously affecting the yield of the medicinal materials and the planting enthusiasm of C. chinensis farmers. As a perennial rhizome medicinal material, C. chinensis requires a large amount of fertiliser, and long-term fertilisation can lead to substantial changes in the quality and functional structure of the soil ecosystem (Luo et al., 2009). Studying the influence of different fertilisation practices on nutrient cycling in farmland and the relationship between fertilisation and the soil environment (Leigh & Johnston, 1994) will be important to improving the yield and quality of medicinal materials and maintaining the health of the soil microenvironment.

Although both fungal and bacterial structures were significantly associated with changes in the rhizosphere soil environment of C. chinensis, the alpha-diversity indices of bacteria showed large differences, while the alpha-diversity indices of fungi showed small differences in the rhizosphere soil of C. chinensis in different growth years (Figs. 5 and 6). In the two planting modes, only three species of fungi had significantly different levels in the rhizosphere soil of C. chinensis in a given growth year (Fig. 7); in contrast, there were large differences in the levels of different bacterial species with 17 bacterial species having significantly different levels between the two planting groups (Fig. 8). Besides their functions and responses to environmental changes, there exist marked differences between fungi and bacteria: fungi establish a more direct connection with plants through hyphae and mycorrhiza, while bacteria are more sensitive to the rhizosphere environment and soil nutrients (Probert et al., 2015). Zhang analysed the variation in the composition and diversity of fungal and bacterial communities in soil at different stages of vegetation restoration after farmland conversion. Both fungal and bacterial communities varied with the succession of plant communities, and the microbial communities under the same plant communities were more similar. However, on the whole, the succession of fungal communities and above-ground plant communities is more synchronized, and fungal communities are more sensitive than bacterial communities to vegetation change (Zhang et al., 2019). In the current study, soil environmental factors, represented by soil physicochemical factors and enzyme activity factors, contributed to 71.49% of the variation in the bacterial community and to 59.18% of the fungal community variation in the rhizosphere soil of C. chinensis. This result also shows that bacteria are more susceptible to the soil environment than fungi (Fig. 13). Therefore, in addition to improving the soil microenvironment through the regulation of nutrient elements, research into the planting modes of C. chinensis, such as rotation and interplanting with other plants, is also necessary to improve the soil microenvironment of C. chinensis.

C. chinensis rhizomes are harvested 5 years after planting and are used in traditional Chinese medicine as source of isoquinoline alkaloids. Seven kinds of alkaloids have been isolated and identified from C. chinensis (Xu et al., 2007). Berberine is a major active component of C. chinensis, which is often used as a way to measure quality with certain levels of berberine required for quality control in Huanglian products (Yan et al., 2008, Xie et al., 2004). The results of an analysis of alkaloid content indicated that in 5-year-old C. chinensis at the harvest stage, berberine levels and total alkaloid levels in the rhizomes were significantly higher in the plants in the natural understory than those planted under a manmade scaffold. These results indicate that compared with planting under a manmade scaffold, planting in the natural understory is more conducive to the accumulation of alkaloids. When planting C. chinensis under a manmade scaffold, the above-ground leaf number, leaf blade weight, petiole weight, and plant height reached their highest values in 4-year-old C. chinensis and became considerably lower in 5-year-old C. chinensis. This decrease is likely because in the 5th year of growth, the roof of the scaffold needs to be removed to increase light intensity and facilitate rhizome growth, commonly known as “drying C. chinensis in the sun.” Our results showed that the shading ratio for 5-year-old C. chinensis was only 13% after the scaffold was removed (Table 1). This exposure to sunlight affected the development of the above-ground part of C. chinensis, causing leaf scorching. Consequently, the indicator values of the above-ground plant parts were much lower in 5-year-old C. chinensis. C. chinensis needs more and more light each year as it grows and develops. For C. chinensis planted in the natural understory, the shading ratio increases and the light intensity decreases as the natural understory continuously grows, which does not work with the increased light needs of C. chinensis. Therefore, when planting C. chinensis as natural understory, the understory shading ratio must be maintained through vegetation thinning and other means to ensure C. chinensis plants get enough light for rhizome growth in the later growth stages.

As mentioned previously, C. chinensis growers in Shizhu have reported that the yield of C. chinensis medicinal materials planted in the natural understory is approximately 20% lower than those planted under a manmade scaffold. However, the above analysis indicates that there were no significant differences in the rhizosphere soil environment of C. chinensis between the two planting modes. Therefore, the yield difference may be caused by sub-optimal field management practices, such as the shading ratio, weeding, and fertilisation in the natural understory mode.

Conclusions

The results of this study found that there were no significant differences in the yield of C. chinensis between the planting modes of natural understory and manmade scaffold under the same field management system, including fertilisation and weeding. Compared to planting under a manmade scaffold, the natural understory planting mode is more conducive to the accumulation of alkaloids in the C. chinensis rhizome. We found that the differences in soil enzyme activity and microbial community structure were mainly between the rhizosphere soil of C. chinensis between different growth years, but there were small differences between the rhizosphere soil of C. chinensis in the same growth year in the different planting modes. We also demonstrated that the physicochemical factors and enzyme activity in the rhizosphere soil of C. chinensis were significantly correlated with variation in microbial community structure. Soil pH, which could also be influenced by the microbial community, was the most important factor influencing variation in both fungal and bacterial community structures. In both planting modes, changes in the nutrient contents in the rhizosphere soil of C. chinensis between different growth years were closely associated with the type and amount of fertilisation applied to C. chinensis. Studying the influence of different fertilisation practices on nutrient cycling in farmland and the relationship between fertilisation and the soil environment will help improve the yield and quality of medicinal materials while maintaining the health of the soil microenvironment.

Supplemental Information

Supplemental Information 1 Changes in enzyme activities in rhizosphere soils of Coptis under both cropping modes chinensis.

Click here for additional data file.

Supplemental Information 2 Dynamic changes in the growth of Coptis chinensis under the planting modes of manmade scaffolding and natural understory.

Click here for additional data file.

Supplemental Information 3 Alkaloid contents in Coptis chinensis rhizomes under the two planting modes.

Click here for additional data file.

Supplemental Information 4 Redundancy analysis on physical-chemical properties and soil dominant fungal phyla.

Click here for additional data file.

Supplemental Information 5 Redundancy analysis on soil physicochemical properties and dominant bacterial phyla.

Click here for additional data file.

Supplemental Information 6 Redundancy analysis on enzyme activity and soil dominant fungal phyla.

Click here for additional data file.

Supplemental Information 7 Redundancy analysis on enzyme activity and soil dominant bacterial phyla.

Click here for additional data file.

We would like to thank AJE for English language editing.

Additional Information and Declarations

Competing Interests

Author Contributions

DNA Deposition

Data Availability

The authors declare that they have no competing interests.

Yu Wang conceived and designed the experiments, prepared figures and/or tables, and approved the final draft.

Yu R. Mo performed the experiments, authored or reviewed drafts of the article, and approved the final draft.

Jun Tan performed the experiments, prepared figures and/or tables, and approved the final draft.

Li X. Wu performed the experiments, authored or reviewed drafts of the article, and approved the final draft.

Yuan Pan analyzed the data, prepared figures and/or tables, and approved the final draft.

Xia D. Chen conceived and designed the experiments, authored or reviewed drafts of the article, and approved the final draft.

The following information was supplied regarding the deposition of DNA sequences:

The group rRNA sequences are available at NCBI: PRJNA779587.

The following information was supplied regarding data availability:

The raw measurements are available in the Supplemental Files.

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
