# Peer review of "Effects of growing Coptis chinensis Franch in the natural understory vs. under a manmade scaffold on its growth, alkaloid contents, and rhizosphere soil microenvironment"

_PeerJ, doi:10.7717/peerj.13676_

## Round 0.1 · original submission · Major Revisions

Please provide a comprehensively revised version addressing the editorial comments and a detailed rebuttal letter.

·

Basic reporting

Dear authors, the manuscript Effects of growing Coptis chinensis Franch in natural understory vs. manmade scaffolding on its growth, alkaloid contents, and rhizosphere soil microenvironment
presents a good contribution in the field, however, it is important that you consider the following:

In general, the spelling and writing of the manuscript must be checked.

Review the format of table 3.

Add some sentences of rationale to the beginning of your abstract.

The changes observed in the alkaloid profile are not discussed.

Experimental design

No comment

Validity of the findings

What do you conclude about the changes that were observed in the alkaloid profile?

·

Basic reporting

a. Some expressions are not concise and need to be refined. For example, line64-72 could be shortened.
b. The authors need to include more background information and reference in “Introduction”. For example, line55-80 should at least discuss previous studies on alkaloids from Coptis chinensis Franch.
c. Table/figure should be referred in your results part. Otherwise, it is hard to follow.
d. Table 4-7: it is better to show RDA results with figures. Tables could be moved to supplementary data.
e. Quality of Figure5-8 should be improved.
f. Some figures could be combined. For example, Figure9 and Figure10 could be combined into one Figure X. Figure X-A refers to Figure9 and Figure X-B refers to Figure10.

Experimental design

a. Reference is needed for some methods: line109-“cutting-ring method”; line112-“five-point method”; line141-“potassium dichromate oxidation-volumetric method”; line142- “Kjeldahl method”, etc.
b. Line164-166: 515F/806R can only target 16S rRNA V4 region. What primers did you use to target all other regions? Primer sequence information should be included.
c. ITS database reference should be mentioned.
d. Line412: RDA analysis should be described in Materials and Methods part.

Validity of the findings

a. Since fertilization was discussed as a key factor, data on fertilization should be provided to support your conclusion.
b. Line530: species or class? Line532: species or class?
c. Line589: soil pH could also be influenced by the microbial community.
d. Alkaloid contents were mentioned in the Title and Abstract part but barely mentioned in Discussion and Conclusion.

Additional comments

A lot of data was provided and analyzed in this study. I recommend the authors to refine the manuscript to make it concise and consistent.

---

## Round 0.2 · Minor Revisions

Please provide a revised version addressing the editorial comments and a detailed rebuttal letter.

·

Basic reporting

1. Spelling and grammar should be carefully checked. e.g. spelling of "Figure"; line452, extra period; line533 understory
2. Abbreviations should be spelled out when it is first used in the article. e.g. line65, TCM.
3. I would suggest the author label the significance level in Table2 like in Table3.
4. Two supplemental tables have the same title.

Experimental design

1. Line176: "ITS1-5F region" should be "ITS1 region". 5F is the primer.
2. Canonical correspondence analysis should be described in methods part.
3. Fertilizer patterns should be moved from discussion to methods part. Also, fertilization rate data is missing.

Validity of the findings

In the discussion part, there are some redundant sentence repeating the results part. I would suggestion the author check carefully and remove the redundant part. e.g. line534-539, the results showed in Table2 was already described.

---

## Round 0.3 · Minor Revisions

The authors have failed to address many typographical and English style requirements for publication in PeerJ. Please use an editorial proofing service to polish your manuscript.

---

## Round 0.4 · accepted · Accept

Thanks, your manuscript is now accepted in PeerJ.

For instance, the Section Editot noted:

> The manuscript is still very verbose. Just one example from the conclusions (first sentence) " The results of this study found that there were no significant differences..", could be written as "We found no significant differences..." or as "The study showed no significant differences...'.